# Bis-Schiff base linkage-triggered highly bright luminescence of gold nanoclusters in aqueous solution at the single-cluster level

Haohua Deng [1,4], Kaiyuan Huang[1,4], Lingfang Xiu[1], Weiming Sun[1], Qiaofeng Yao[2,3], Xiangyu Fang[1], Xin Huang[1], Hamada A. A. Noreldeen[1], Huaping Peng[1], Jianping Xie [2,3✉] & Wei Chen[1✉]

Metal nanoclusters (NCs) have been developed as a new class of luminescent nanomaterials with potential applications in various fields. However, for most of the metal NCs reported so far, the relatively low photoluminescence quantum yield (QY) in aqueous solution hinders their applications. Here, we describe the utilization of bis-Schiff base linkages to restrict intramolecular motion of surface motifs at the single-cluster level. Based on $Au_{22}(SG)_{18}$ (SG: glutathione) NCs, an intracluster cross-linking system was constructed with 2,6-pyridinedicarboxaldehyde (PDA), and water-soluble gold NCs with luminescence QY up to 48% were obtained. The proposed approach for achieving high emission efficiency can be extended to other luminescent gold NCs with core-shell structure. Our results also show that the content of surface-bound Au(I)-SG complexes has a significant impact on the PDA-induced luminescence enhancement, and a high ratio of Au(I)-SG will be beneficial to increasing the photoluminescence intensity of gold NCs.

[1] Higher Educational Key Laboratory for Nano Biomedical Technology of Fujian Province, Department of Pharmaceutical Analysis, Fujian Medical University, 350004 Fuzhou, China. [2] Department of Chemical and Biomolecular Engineering, National University of Singapore, 4 Engineering Drive 4, Singapore 117585, Singapore. [3] Joint School of National University of Singapore and Tianjin University, International Campus of Tianjin University, Binhai New City, 350207 Fuzhou, China. [4] These authors contributed equally: Haohua Deng, Kaiyuan Huang. ✉email: chexiej@nus.edu.sg; weichen@fjmu.edu.cn

Over the past few decades, metal nanoclusters (NCs) have emerged as promising functional materials and have attracted considerable attention owing to their unique structural, optical, electrical, magnetic, and catalytic properties[1–5]. Among these properties, visible to near-infrared photoluminescence (PL) is one of the most appealing due to the extensive use of PL-based techniques in chemical sensing, biological imaging, and light-emitting devices[6–11]. However, the relatively weak luminescence of metal NCs in comparison with organic dyes, fluorescent proteins, and semiconductor quantum dots has been a non-negligible obstacle to their applicability[12,13]. As a consequence, an increasing amount of effort in current research on luminescent metal NCs is being dedicated to developing effective strategies to improve their PL quantum yields (QYs).

Metal NCs typically have a core-shell structure that composes of an inner metal kernel and outer metal-ligand motifs[14]. So far, it has been widely accepted that the luminescence property of metal NCs is closely associated with the metal-ligand motifs on NC surface, and the restriction of intramolecular motion (RIM) of these surface-bound complexes is an efficient way to enhance the emission efficiency of metal NCs. The activation of the RIM process can decrease the energy loss of photoexcited states through nonradiative relaxation, and the energy released through the radiative transition would increase correspondingly[15]. Hitherto, various methods have been used to trigger the RIM process of metal NCs, which can be mainly classified into four strategies: (1) solvent- or cation-induced aggregation and self-assembly[16–22]; (2) enhancing rigidity via binding with bulky groups[23]; (3) host-guest interactions[24–26]; and (4) spatial confinement[27]. Despite such prosperous achievements, it remains challenging to obtain ultrabright metal NCs in an aqueous solution because most of the above-mentioned systems have some or all of the following deficiencies: (1) they usually generate the large size of architectures with poor stability and controllability; (2) the resulting products cannot be well dispersed in water, which is unfavorable to their further employment in the biomedical field; and (3) surface engineering is often required, leading to a complicated design procedure with poor universality.

Dialdehydes are a family of chemical compounds containing two aldehyde (–CHO) groups, which have been frequently adopted as diverse building blocks for the construction of nanoscale architectures (e.g., metal-organic frameworks, covalent organic frameworks, nanocellulose, and nanotubes) with high precision and good controllability[28–31]. Dialdehydes can react with amino groups to form imine (–CH=N–) bonds to induce cross-linking of the components, which enables the fabricated

materials to own the advantages of a Schiff base: (1) high stability compared to the noncovalent interaction, (2) mild reaction condition with water as the sole byproduct, (3) high reaction rate in aqueous solution, and (4) introduction of rigid structure which would benefit for luminescence enhancement[32]. Cross-linking by dialdehydes can not only improve the performances and properties of materials but also provide new opportunities with regard to the design and practical application of these materials[33–35]. Nevertheless, this cross-linking strategy has not been utilized to attain highly emissive metal NCs. The dialdehyde-mediated cross-linking strategy is particularly attractive for water-soluble metal NCs because amino groups are common functional groups on their surface, which is very feasible for the formation of Schiff bases.

Based on the background mentioned above, herein, we report a simple and straightforward way to improve the luminescence efficiency of metal NCs effectively. Water-soluble glutathione (SG)-stabilized gold NCs are employed to probe the effect of interactions between Au(I)-SG motifs and dialdehydes on cluster luminescence in the scheme of activating the RIM. It is found that 2,6-pyridinedicarboxaldehyde (PDA) can serve as an excellent building block to construct an intracluster cross-linking system (i.e., cross-linking at the single-cluster level), which can obviously boost the luminescence intensity of gold NCs. Time-resolved luminescence measurements identify a longer decay lifetime after conjugating with PDA, manifesting that the nonradiative pathway of the luminescent gold NCs is suppressed. To clearly understand the PL enhancement, ultrafast transient absorption spectroscopy is exploited to monitor the electron dynamics of gold NCs before and after the introduction of PDA. Further experiments show that the emission enhancement of gold NCs triggered by PDA displays a strong dependency on the Au(I)-SG content.

## Results

**Selection and characterization of gold NCs.** The $Au_{22}(SR)_{18}$ (SR: thiolate) NCs, which were firstly reported by Xie et al.[36], have shown a great application prospect in the fields of bioimaging, solar reduction, and mimicking the pathogenic invasion because of their PL property, light-harvest ability, and biocompatibility[37–39]. Crystal investigations and extended X-ray absorption fine structure measurements revealed that $Au_{22}(SR)_{18}$ clusters possess a bitetrahedral $Au_7$ core surrounded by one $Au_6(SR)_6$ ring motif and bridged by three bidentate $Au_3(SR)_4$ (Fig. 1a)[40–43]. Such a unique feature of $Au_{22}(SR)_{18}$ clusters renders them a large degree of structural flexibility, which can benefit their further PL efficiency improvement by suppressing the relaxation of Au(I)-SR motifs. Considering the

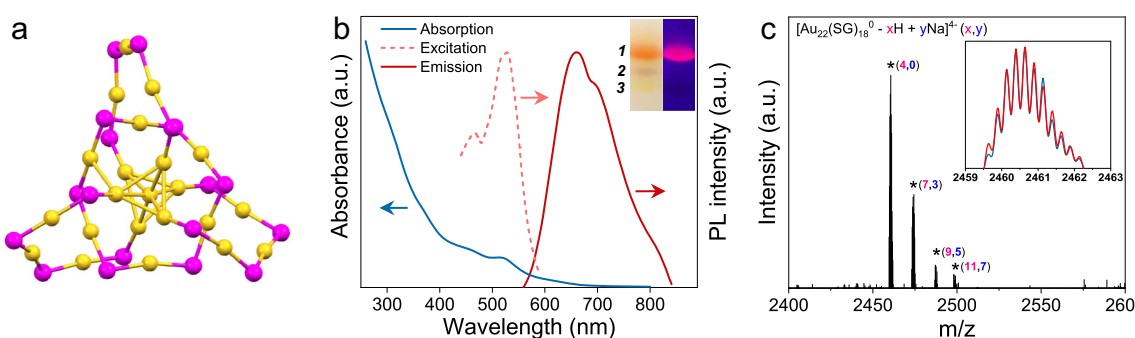

**Fig. 1 Characterization of $Au_{22}(SG)_{18}$ nanoclusters. a** The Au-S framework of $Au_{22}(SR)_{18}$ nanoclusters (NCs) (drawn according to the reported structure[43]). The yellow spheres represent Au atoms, and the purple spheres represent S atoms. **b** UV-vis absorption, photoexcitation, and photoemission spectra of the purified $Au_{22}(SG)_{18}$ NCs. Inset: Digital photos of the PAGE gels of gold NCs under visible (left) and UV (right) light. Bands 1–3 correspond to $Au_{22}(SG)_{18}$, $Au_{18}(SG)_{14}$, and $Au_{15}(SG)_{13}$, respectively. a.u. arbitrary units. **c** Electrospray ionization (ESI) mass spectrum of $Au_{22}(SG)_{18}$ NCs. Inset shows the experimental (blue line) and simulated (red line) isotope patterns of $[Au_{22}(SG)_{18}{}^0–4H]^{4-}$. a.u. arbitrary units.

abundant amino groups on the surface of $Au_{22}(SG)_{18}$ (SG: glutathione) clusters, which are conducive to the formation of bis-Schiff base with dialdehydes, we selected $Au_{22}(SG)_{18}$ clusters as a model NC to illustrate the feasibility of our proposed dialdehydes-mediated cross-linking strategy to activate the RIM process and boost emission efficiency of luminescent metal NCs.

The $Au_{22}(SG)_{18}$ NCs were prepared by following a reported method with slight modifications[23,44]. Three kinds of gold NCs, namely $Au_{15}(SG)_{13}$, $Au_{18}(SG)_{14}$, and $Au_{22}(SG)_{18}$, were identified by using a native polyacrylamide gel electrophoresis (PAGE, 30%) (Fig. 1b inset), corresponding to the results of previous works[23,36]. Highly pure $Au_{22}(SG)_{18}$ NCs were then isolated from red-emitting fraction of separating gel and redissolved in water. The obtained $Au_{22}(SG)_{18}$ NCs with molecular-level purity display two characteristic UV-vis absorption peaks at 460 and 515 nm, respectively, and an emission peak at about 660 nm (Fig. 1b). The resulting photoexcitation spectrum showed two peaks at 465 and 520 nm in the visible region, which nicely matched the prominent maxima in the absorption spectra. A series of peaks with a spacing of 0.25 that represented $Au_{22}(SG)_{18}$ ions containing different numbers of $Na^+$ ions ($n = 0$–7) were clarified at $m/z$ 2400–2600 Da in the negative-mode electrospray ionization mass spectrum (ESI-MS) (Fig. 1c). The zoom-in mass spectrum of the most intense peak at $m/z$ ~2460 Da equates well with the simulated isotopic pattern of $[Au_{22}(SG)_{18}{}^0{-}4H]^{4-}$ (Fig. 1c inset). The absolute luminescence QY of the as-isolated $Au_{22}(SG)_{18}$ NCs was determined to be 4.6% (pH = 11).

**Rational design of dialdehydes to activate the RIM process.** Since stability and reaction rate of imine linkers are important parameters affecting the activation of the RIM process of the Au(I)-SG motifs on NC surface, we initiate theoretical studies to examine bis-Schiff base formation reaction of a series of commonly used dialdehydes in an aqueous solution. The tested dialdehydes are glutaraldehyde (GA), m-phthalaldehyde (mPA), furan-2,5-dicarbaldehyde (DFF), and 2,6-pyridinedicarboxaldehyde (PDA). The results of thermodynamics simulations displayed that the free energy change value (ΔG) along with the equilibrium constant of bis-Schiff base reaction between PDA and α-amino acid ($RCHNH_2COOH$, R=H) is the highest among all dialdehydes studied (Table 1), implying that the imine linkage formed between PDA and SG is the most stable one. Furthermore, we calculated the activating energy of the bis-Schiff base reaction of dialdehydes (Fig. 2a, b and Supplementary Figs. 1–4), and Cartesian coordinates for the calculated structures were provided in Supplementary Data 1. The lowest energy barrier was observed when PDA undergoes bis-Schiff base reaction in the aqueous solution, which suggests a high reaction rate (Table 1). To confirm the results of theoretical calculations, these selected dialdehydes were introduced into $Au_{22}(SG)_{18}$ cluster solution, and time-dependent luminescence intensities of the

mixtures were measured. As depicted in Fig. 2c, d, the brightest luminescence intensity with the fastest growth was achieved in the case of PDA-$Au_{22}(SG)_{18}$ NCs, which agrees well with the corresponding theoretical results. Hereafter, PDA was selected to cross-link Au(I)-SG motifs to activate the RIM process of $Au_{22}(SG)_{18}$ NCs.

Next, a series of experiments were carried out to evaluate the interaction between $Au_{22}(SG)_{18}$ NCs and PDA. As shown in Fig. 3a, the FT-IR spectrum of PDA-$Au_{22}(SG)_{18}$ NCs displays a typical imine stretching vibration at $1660\,cm^{-1}$, unambiguously indicating the occurrence of Schiff base reactions between the –CHO groups of PDA and –$NH_2$ functionalities on the surface of $Au_{22}(SG)_{18}$ NCs[45]. The formation of imine linkages can be further evidenced by $^1H$ NMR measurements and UV absorption spectra. It can be seen from Fig. 3b that a broad peak at ~10.0 ppm corresponds to the –CHO group and a sharp peak at 8.6 ppm that belongs to the proton of the –CH=N– group were detected in the PDA-$Au_{22}(SG)_{18}$ clusters[46]. Conversely, no obvious –CH=N– signal was found in mPA–$Au_{22}(SG)_{18}$ and GA–$Au_{22}(SG)_{18}$ NCs (Supplementary Fig. 5), confirming our proposed mechanism of the bis-Schiff base linkage-triggered emission enhancement by activation of RIM. The absorption peak of PDA caused by π–π* transition undergoes a distinct redshift (275→290 nm) upon combining with $Au_{22}(SG)_{18}$ NCs (Supplementary Fig. 6), which comes from the formation of a greater π bond conjugate system. Besides, in the spectrum of PDA-$Au_{22}(SG)_{18}$ NCs, two new weak absorption peaks appear at 310 and 323 nm, which can be assigned to the n–π* electron transition of N atoms of pyridine ring and Schiff base, respectively[47]. Since SG contains only one amino group and each PDA can react with two amino groups to produce bis-Schiff base compound, the binding number of PDA on each $Au_{22}(SG)_{18}$ cluster is supposed to be 9. This assumption can be confirmed by the PL titration experiments. As shown in Fig. 3c, the plot of the PDA concentration-dependent luminescence intensity enhancement for $Au_{22}(SG)_{18}$ NCs reveals that there are on average 9 PDA conjugated to one $Au_{22}(SG)_{18}$ cluster (PDA: $Au_{22}(SG)_{18}$ = 273:30 = 9.1). A post reaction with an amino-reactive reagent suggested that almost all of the amino groups on the surface of $Au_{22}(SG)_{18}$ NCs interacted with PDA (Supplementary Fig. 7). The transmission electron microscope (TEM) images demonstrated that the $Au_{22}(SG)_{18}$ NCs remain individual in the presence of PDA (Supplementary Fig. 8) and other dialdehydes (Supplementary Fig. 9). Dynamic light scattering (DLS) measurements also testified that no large-scale aggregates or assemblies of $Au_{22}(SG)_{18}$ NCs are generated after the interaction with PDA (Supplementary Fig. 10). In addition, the PAGE gel (Supplementary Fig. 11) clearly shows that both $Au_{22}(SG)_{18}$ and PDA-$Au_{22}(SG)_{18}$ NCs exhibit a similar separating band. Taken together, these data from TEM, DLS, and PAGE analysis verified the good dispersibility of PDA-$Au_{22}(SG)_{18}$ NCs and excluded the possibility of PL efficiency enhancement caused by cross-linking among clusters.

**Optical properties of PDA-$Au_{22}(SG)_{18}$ NCs.** To unveil the underlying mechanism of the intracluster cross-link-enhanced emission by activation of RIM at the single-cluster level, optical properties of the formed PDA-$Au_{22}(SG)_{18}$ NCs were studied. As shown in Supplementary Fig. 12, absorbance spectra of $Au_{22}(SG)_{18}$ and PDA-$Au_{22}(SG)_{18}$ NCs in the visible region are almost identical, revealing the character of core electron transition ($S_0$–$S_1$) in these NCs remains unaltered. On contrary, the luminescent intensity was strengthened considerably after being reacted with PDA (Fig. 3d), and this intensity was found to be stable in the pH range of 9–12 (Supplementary Fig. 13).

**Table 1 Calculated parameters for the formation of bis-Schiff base.**

| Dialdehyde | $-\Delta G$ (kcal/mol)[a] | K ($L^2 \cdot mol^{-2}$)[b] | $Ea_{max}$ (kcal/mol)[c] |
|---|---|---|---|
| GA | 4.54 | $2.14 \times 10^3$ | 37.1 |
| mPA | 6.38 | $4.81 \times 10^4$ | 35.2 |
| DFF | 9.02 | $4.16 \times 10^6$ | 36.9 |
| PDA | 9.38 | $7.57 \times 10^6$ | 34.9 |

[a]ΔG = free energy change.
[b]K = equilibrium constant.
[c]$Ea_{max}$ = maximum activating energy.

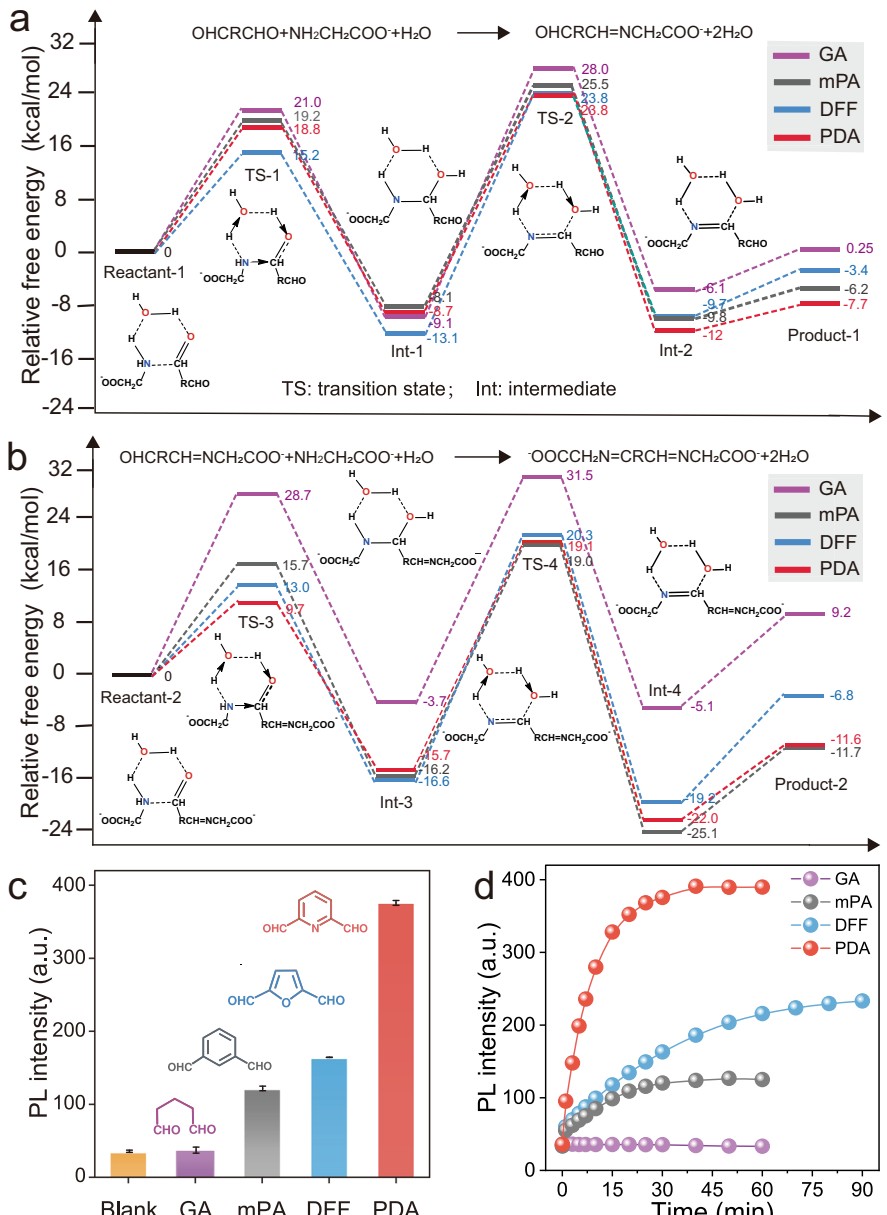

**Fig. 2 Theoretical and experimental investigations of the bis-Schiff base formation reaction. a** Calculated free energy profiles for the formation processes of mono-Schiff bases. GA glutaraldehyde, mPA m-phthalaldehyde, DFF furan-2,5-dicarbaldehyde, PDA 2,6-pyridinedicarboxaldehyde. **b** Calculated free energy profiles for the formation processes of bis-Schiff bases. Energies are calculated using B3LYP/6-311+G(d, p) method in water. **c** Comparison of the luminescence intensity enhancement effects of different dialdehydes on $Au_{22}(SG)_{18}$ NCs. Inset shows the chemical structures of GA, mPA, DFF, and PDA. Error bars represent standard deviation over three independent measurements. a.u. arbitrary units. **d** Time-dependent luminescence intensities of $Au_{22}(SG)_{18}$ NCs in the presence of different dialdehydes. a.u. arbitrary units.

The intensity increased over 11-fold, and a luminescence QY of $48 \pm 1.4\%$ was achieved for PDA-$Au_{22}(SG)_{18}$ NCs in an aqueous solution at room temperature (Supplementary Fig. 14). In addition, a blue shift of the emission peak ($\lambda_{em}$) from 660 to 640 nm with a decrease of Stokes shift (about 0.06 eV) was noticed, which is a typical characteristic of RIM owing to the suppression of the nonradiative decay pathway of the luminescent surface state[48]. A control test demonstrated that PDA and its complex with SG did not show obvious emission under the same experimental conditions (Supplementary Fig. 15), proving that the observed PL enhancement indeed stems from the intensive interactions between PDA and $Au_{22}(SG)_{18}$ NCs. Moreover, it is noteworthy that no apparent luminescence enhancement was discovered

when using 2-pyridinecarboxaldehyde instead of PDA (Supplementary Fig. 16). This result confirms that the bis-Schiff base linkages are essential for the observation of highly bright luminescence from PDA-$Au_{22}(SG)_{18}$ NCs because they can significantly restrain the movement of Au(I)-SG motifs (Fig. 3e). To gain insight into the details of the change in the decay pathway, we measured the luminescence lifetimes of $Au_{22}(SG)_{18}$ and PDA-$Au_{22}(SG)_{18}$ NCs (Fig. 3f). It clearly shows that the luminescence lifetime ($\tau$) of PDA-$Au_{22}(SG)_{18}$ NCs is longer than that of $Au_{22}(SG)_{18}$ NCs (9.3 µs vs 5.4 µs) (Supplementary Table 1). The microsecond-scale lifetimes implied that the emissions from both clusters were derived from the Au(I)-SG motifs on the NC shell involving triplet excited states, namely, phosphorescence[23,36].

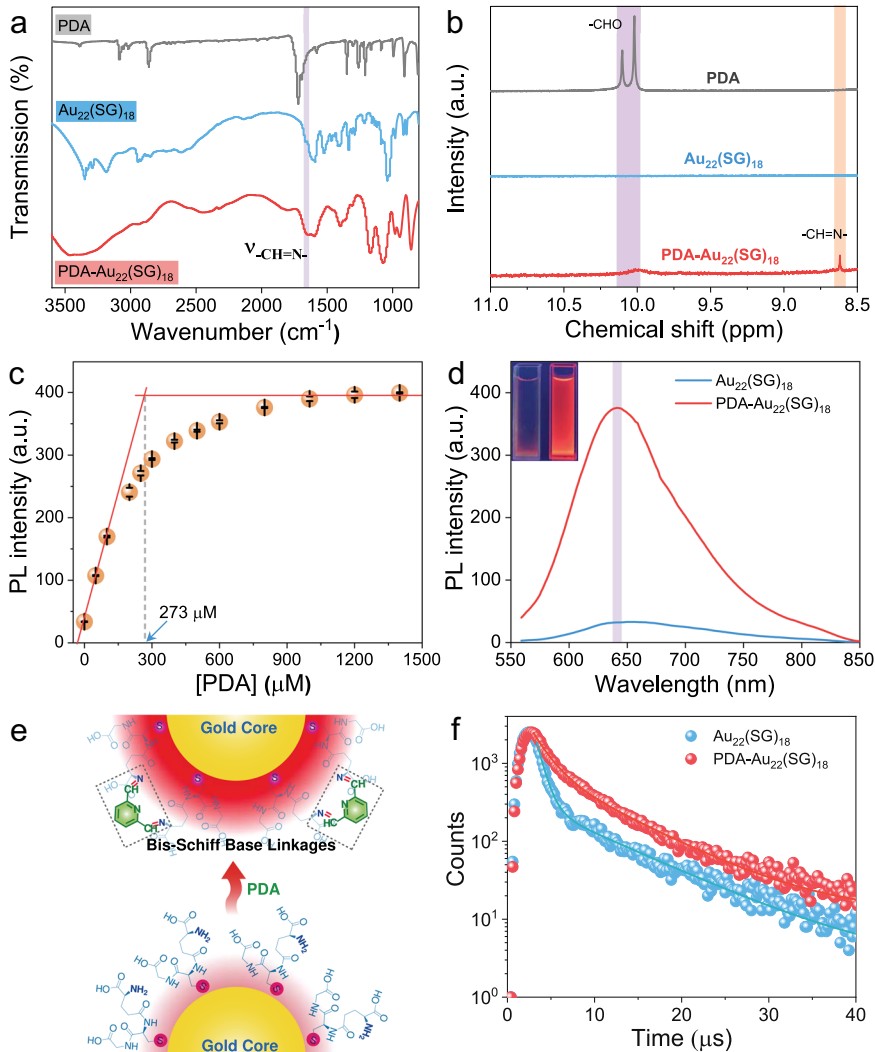

**Fig. 3 Interaction between Au₂₂(SG)₁₈ NCs and PDA. a** FT-IR spectra of PDA, Au₂₂(SG)₁₈, and PDA-Au₂₂(SG)₁₈ NCs. The purple area represents an imine stretching vibration region. **b** ¹H NMR spectra of PDA, Au₂₂(SG)₁₈ NCs, and PDA-Au₂₂(SG)₁₈ NCs. The purple and orange area correspond to the characteristic signals of –CHO and –CH=N–, respectively. a.u. arbitrary units. **c** PDA concentration-dependent luminescence intensity enhancement for Au₂₂(SG)₁₈ NCs. The solid red lines are tangents of initial and final stages of the titration plots and the dotted line is a guide for intersection point of the tangent lines. The concentration of Au₂₂(SG)₁₈ NCs was 30 μM. Error bars represent standard deviation over three independent measurements. a.u. arbitrary units. **d** Photoemission spectra of Au₂₂(SG)₁₈ and PDA-Au₂₂(SG)₁₈ NCs. The purple area shows the maximum emission of PDA-Au₂₂(SG)₁₈ NCs at a shorter-wavelength region. Inset shows the digital photos of Au₂₂(SG)₁₈ (left) and PDA-Au₂₂(SG)₁₈ (right) NCs under UV light. The excitation wavelength was 515 nm. a.u. arbitrary units. **e** Schematic of the correlation of emission intensity with the binding PDA. The binding of PDA on the Au₂₂(SG)₁₈ surface can induce intramolecular cross-linking of ligands by formation of bis-Schiff linkages, which gives rise to PL enhancement. The red shadings represent PL intensities of Au₂₂(SG)₁₈ NCs before and after the addition of PDA. **f** Comparison of the luminescence decay of Au₂₂(SG)₁₈ and PDA-Au₂₂(SG)₁₈ NCs.

On the basis of the luminescence QY and lifetime data, we further solved the radiative decay rate $k_R$ and nonradiative decay rate $k_{nR}$ of these two NCs using the following equations:[49]

$$QY = \frac{k_R}{k_R + k_{nR}} \quad (1)$$

$$\tau = \frac{1}{k_R + k_{nR}} \quad (2)$$

One can see that the luminescence QY is highly dependent on the competition between $k_R$ and $k_{nR}$. Either reducing $k_{nR}$ or increasing $k_R$ would improve the luminescence QY. As listed in Supplementary Table 1, the $k_R$ of PDA-Au₂₂(SG)₁₈ NCs dramatically increased while the $k_{nR}$ suffered a remarkable decrease. This trend obeys the energy gap law, where $k_R$ increases

and $k_{nR}$ decreases with the increase of the optical energy gap ($E_{opt}$, $E_{opt} = 1240/\lambda_{em}$)[50]. Thus, we can conclude that the PDA-Au₂₂(SG)₁₈ NCs with such a high luminescence QY are mainly attributed to the vigorously suppressed nonradiative loss and the greatly increased radiative recombination rate, which is directly related to the formation of bis-Schiff base linkages and confined motion of Au(I)-SG staple motifs.

**Unraveling the electronic dynamics of PDA-Au₂₂(SG)₁₈ NCs.**
To reveal the origin of the suppressed nonradiative decay and increased radiative decay pathways, femtosecond transient absorption (fs-TA) measurements were performed to determine the electronic dynamics of Au₂₂(SG)₁₈ and PDA-Au₂₂(SG)₁₈ NCs. Upon photoexcitation with a 400 nm laser pulse, apparent excited-state absorption (ESA) can be identified in the fs-TA

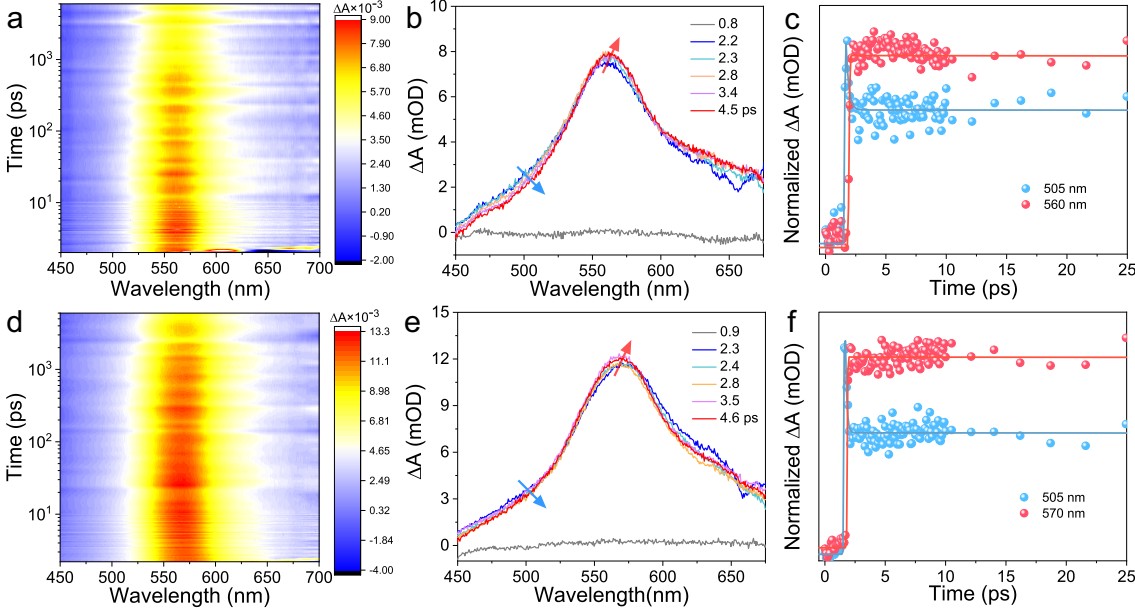

**Fig. 4 Femtosecond transient absorption measurements. a** Two-dimensional femtosecond transient absorption (fs-TA) spectra of $Au_{22}(SG)_{18}$ NCs. **b** fs-TA spectra of $Au_{22}(SG)_{18}$ NCs at short time delays. **c** Kinetic traces and fitting lines of $Au_{22}(SG)_{18}$ NCs at 505 and 560 nm. **d** Two-dimensional fs-TA spectra of PDA-$Au_{22}(SG)_{18}$ NCs. **e** fs-TA spectra of PDA-$Au_{22}(SG)_{18}$ NCs at short time delays. **f** Kinetic traces and fitting lines of PDA-$Au_{22}(SG)_{18}$ NCs at 505 and 570 nm.

spectra of $Au_{22}(SG)_{18}$ NCs (Fig. 4a). A growth of 560 nm ESA accompanied by the decay of 505 nm ESA indicates the core-shell transitions (Fig. 4b). Specifically, the analysis of dynamics traces of 505 nm ESA gives a short lifetime with 220 fs (Fig. 4c, blue line), which can be assigned to the internal conversion (IC) of gold core from a higher singlet state ($S_n$) to the lowest singlet state ($S_1$)[51]. The ESA at 560 nm gives a rise with a time constant of 300 fs (Fig. 4c, red line), which suggests an ultrafast core-shell transition, i.e., core-to-shell intersystem crossing (ISC) from $S_1$ to the lowest triplet state ($T_1$), owing to the small energy gap (<0.1 eV) between $S_1$ state and $T_1$ state ($\Delta E_{S_1-T_1}$) of $Au_{22}(SG)_{18}$ NCs[42,52,53]. Intriguingly, after treatment with PDA, the shell state absorption of $Au_{22}(SG)_{18}$ NCs red-shifted to 570 nm with a decrease of energy gap (0.04 eV) (Fig. 4d), which is in accordance with the small decrease of Stokes shift as indicated by their luminescence spectra (please refer to Fig. 3d). Meanwhile, the fs-TA analysis of PDA-$Au_{22}(SG)_{18}$ NCs also displays decay at 505 nm that can be assigned to the intracore state of $Au_{22}(SG)_{18}$ NCs (Fig. 4e). The fitting decay profile of ESA at 505 nm shows a shorter decay time of 120 fs, suggesting a more rapid intracore state relaxation (Fig. 4f, blue line). The growth of ESA at 570 nm happens at a very short time scale within 150 fs (Fig. 4f, red line). Since energy transfer and photoinduced electron transfer (pyridine anion radical absorption was not identified) are unfavorable between $Au_{22}(SG)_{18}$ cluster and PDA (Supplementary Fig. 17)[54,55], the faster relaxation process can be attributed to the promoted core-shell transition in PDA-$Au_{22}(SG)_{18}$ NCs. That may be due to a reduction in interaction distance between kernel and shell after the formation of bis-Schiff base because core-shell transition in noble NCs via a charge recombination mechanism usually displays donor-acceptor distance-dependent electron transfer rate. Besides, as stated before, a blue-shifted emission peak was found when $Au_{22}(SG)_{18}$ cluster interacted with PDA, which means a higher $T_1$ energy level and a smaller $\Delta E_{S_1-T_1}$ in PDA-$Au_{22}(SG)_{18}$ NCs compared to $Au_{22}(SG)_{18}$ NCs. Thus, the observed higher ISC rate for PDA-$Au_{22}(SG)_{18}$ NCs is also expected according to the rule $k_{ISC} \propto \frac{1}{(\Delta E_{S_1-T_1})^2}$ [56]. The higher $T_1$ state in PDA-$Au_{22}(SG)_{18}$

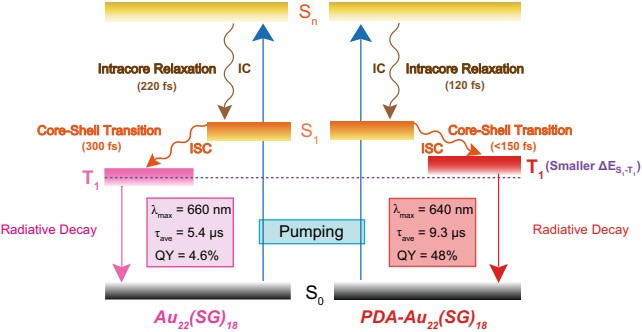

**Fig. 5 Schematic diagram illustrating the excited-state dynamics of $Au_{22}(SG)_{18}$ and PDA-$Au_{22}(SG)_{18}$ NCs.** Arrows denote the transitions between different electronic states. $S_0$ ground state, $S_n$ high singlet state, $S_1$ lowest singlet state, $T_1$ lowest triplet state, IC internal conversion, ISC intersystem crossing, $\Delta E_{S_1-T_1}$ energy gap between $S_1$ and $T_1$ states, $\lambda_{max}$ maximum emission wavelength, $\tau_{ave}$ average luminescence lifetime, QY luminescence quantum yield.

NCs can be further supported by the detection of NC-photosensitized singlet oxygen formation using electron spin resonance spectroscopy. As can be seen from Supplementary Fig. 18, an increase in singlet oxygen ($^1O_2$) signal was found in the system of PDA-$Au_{22}(SG)_{18}$ NCs, which originated from their high $T_1$ energy level and large driving force for triplet–triplet energy transfer[57]. Combined with the measurements of steady and time-resolved absorption, and luminescence spectra, we tried to give a rational explanation to the observed PL enhancement of $Au_{22}(SG)_{18}$ NCs induced by PDA, which can be described as follows (Fig. 5). When ground state ($S_0$) electrons of $Au_{22}(SG)_{18}$ are pumped, they undergo a fast decay into $T_1$ state by the IC process of the core state from $S_n$ to $S_1$ and the ISC process of the core-shell state from $S_1$ to $T_1$, subsequently, radiative decay occurs in the $T_1$ state. After the introduction of PDA, the formed bis-Schiff base linkages make the surface-bound Au(I)-SG

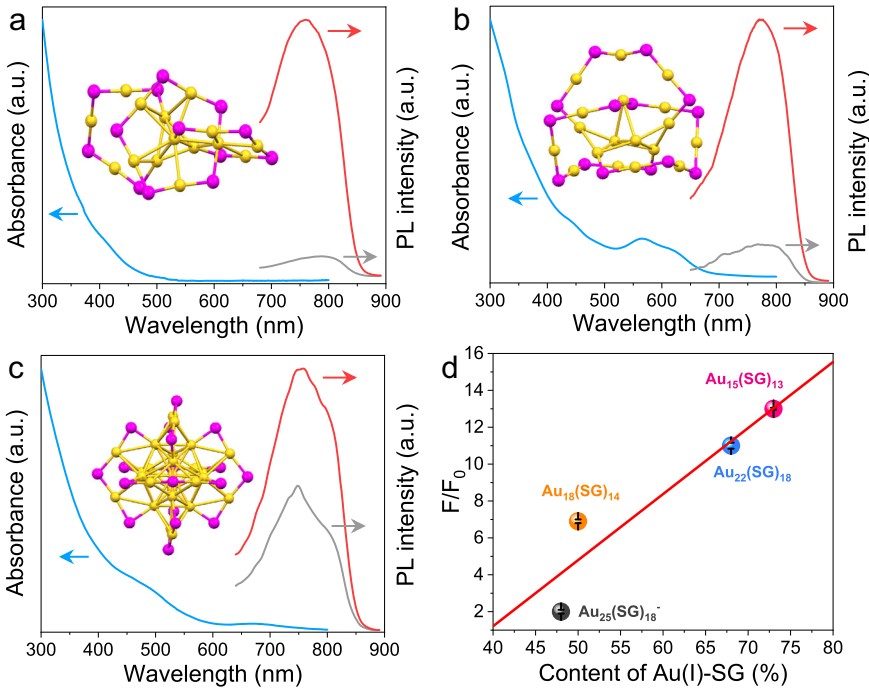

**Fig. 6 PDA-triggered luminescence enhancement of other types of gold NCs. a** UV-vis absorption spectrum of $Au_{15}(SG)_{13}$ (blue line), and photoemission spectra of $Au_{15}(SG)_{13}$ (gray line) and PDA-$Au_{15}(SG)_{13}$ (red line) NCs. Inset shows the Au-S framework of $Au_{15}(SR)_{13}$ (drawn according to the reported structure[60]). The yellow spheres represent Au atoms, and the purple spheres represent S atoms. a.u. arbitrary units. **b** UV-vis absorption spectrum of $Au_{18}(SG)_{14}$ (blue line), and photoemission spectra of $Au_{18}(SG)_{14}$ (gray line) and PDA-$Au_{18}(SG)_{14}$ (red line) NCs. Inset shows the Au-S framework of $Au_{18}(SR)_{14}$ (drawn according to the reported structure[61]). a.u. arbitrary units. **c** UV-vis absorption spectrum of $Au_{25}(SG)_{18}^-$ (blue line), and photoemission spectra of $Au_{25}(SG)_{18}^-$ (gray line) and PDA-$Au_{25}(SG)_{18}^-$ (red line) NCs. Inset shows the Au-S framework of $Au_{25}(SR)_{18}^-$ (drawn according to the reported structure[62]). a.u. arbitrary units. **d** Effect of the content of surface-bound Au(I)-SG complexes on the PDA-induced luminescence enhancement of gold NCs. $F_O$ and $F$ represent the luminescence intensity of gold NCs at their maximum emission wavelengths in the absence and presence of PDA, respectively. Error bars represent standard deviation over three independent measurements.

complexes less flexible, which might shorten the interaction distance between $Au_7$ core and Au(I)-SG motifs of $Au_{22}(SG)_{18}$ NCs and yield a higher $T_1$ state with a smaller $\Delta E_{S_1-T_1}$. As a result, the ISC process in PDA-$Au_{22}(SG)_{18}$ NCs becomes much faster than that of $Au_{22}(SG)_{18}$ NCs, leading to significant suppression of nonradiative energy loss and a dramatic increase in radiative recombination rate. Therefore, the electrons at the $T_1$ states emit a bright shorter-wavelength light with a high luminescence QY.

**Generality of the proposed strategy**. The proposed strategy for luminescence enhancement by PDA-mediated cross-linking of Au(I)-SG motifs was found to be suitable for other gold NCs with core-shell structures. Other three kinds of gold NCs including $Au_{15}(SG)_{13}$, $Au_{18}(SG)_{14}$, and $Au_{25}(SG)_{18}^-$ were synthesized based on previously reported methods and purified by native PAGE (the corresponding ESI mass spectra were shown in Supplementary Fig. 19)[23,58], and the cross-linking procedure for these clusters was similar to that of PDA-$Au_{22}(SG)_{18}$. UV-vis absorption spectra indicate that all these gold NCs show their own characteristic absorption peaks (370 and 410 nm for $Au_{15}(SG)_{13}$, 570 and 620 nm for $Au_{18}(SG)_{14}$, and 680 nm for $Au_{25}(SG)_{18}^-$ (blue lines, Fig. 6a–c), which is consistent with the results in an earlier study[59]. As shown in Fig. 6a–c, weak emission is observed from these clusters (gray lines); however, after treatment with PDA, their luminescence intensity can be dramatically enhanced (red lines). Note that all these gold NCs display a blue-shift in the emission spectrum when PDA is present, suggesting that a RIM mechanism is typically involved in the observed luminescence increasement. Interestingly, we further found that the luminescence enhancing performance of PDA is strongly

related to the content of Au(I)-SG complexes on the NC shell. Previous investigations have demonstrated that the $Au_{15}(SR)_{13}$, $Au_{22}(SR)_{18}$, $Au_{18}(SR)_{14}$, and $Au_{25}(SR)_{18}^-$ NCs possess a tetrahedral $Au_4$ core (Fig. 6a, inset)[60], a bitetrahedral $Au_7$ core (Fig. 1a), a dioctahedral $Au_9$ core (Fig. 6b, inset)[61], and an icosahedral $Au_{13}$ core (Fig. 6c, inset)[62], respectively, suggesting an increase in the size of Au(0) core and a decline in the content of surface-capped Au(I)-SG complex for these NCs. From Fig. 6d, it can be clearly seen that the luminescence enhancing efficiency induced by PDA increases with increased Au(I)-SG content, following the order of $Au_{15}(SG)_{13}$ (an increase of about 13 times) > $Au_{22}(SG)_{18}$ (ca. 11 times) > $Au_{18}(SG)_{14}$ (ca. 7 times) > $Au_{25}(SG)_{18}^-$ (ca. two times). Such a tendency is reasonable because more excited-state energy will be lost via nonradiative decay at the surface of gold NCs with a higher percentage of Au(I)-SG, and thus the activation of the RIM process of these surface-bound complexes can boost the NC's luminescence more effectively in this situation. In addition, significant luminescence improvements were also observed for other ligand systems containing –$NH_2$ groups (γ-Glu-Cys, Gly-Cys-Gly, and L-penicillamine) (Supplementary Figs. 20a-20c). Notably, because there is no –$NH_2$ group to achieve imine bond formation, N-acetyl-L-cysteine-protected gold NCs did not show any obvious luminescence improvement with the addition of PDA (Supplementary Fig. 20d). These results demonstrate the good generality of the as-developed luminescence enhancement strategy.

**Discussion**

In summary, we accomplished the enhancement of metal NC luminescence in the aqueous solution by activation of the RIM

process at the single-cluster level through a dialdehyde-mediated intracluster cross-linking strategy. The influence of interactions between Au(I)-SG motifs and dialdehydes on the cluster luminescence was investigated by employing $Au_{22}(SG)_{18}$ NCs as a model NC, leveraging on their good structural flexibility and amino groups-enriched surface. It has been shown that among all the selected dialdehydes, PDA has the highest luminescence enhancement ability for $Au_{22}(SG)_{18}$ NCs. The bis-Schiff base linkages formed between PDA and SG ligands can evidently suppress flexibility of the Au(I)-SG motifs on the surface of $Au_{22}(SG)_{18}$ NCs, leading to a remarkable decrease in nonradiative decay rate and an obvious increase in the radiative decay rate. Femtosecond transient absorption measurements revealed that the internal conversion and intersystem crossing processes of $Au_{22}(SG)_{18}$ NCs were facilitated after being interacted with PDA. Accordingly, enhanced photoemission from PDA-$Au_{22}(SG)_{18}$ NCs with a luminescence QY as high as 48% was realized through the vigorous suppression of nonradiative energy loss and the great increase in the radiative recombination rate. Furthermore, this investigative approach for improving the luminescence efficiency can be extended to other luminescent gold NCs with a core-shell structure. Results showed that a higher content of Au(I)-SG complexes on the NC shell can produce a better luminescent enhancement effect. This study offers an in-depth understanding of the structure-luminescence relationship of metal NCs and provides a promising way for the rational design of highly emissive metal NCs for their subsequent use in biomedical imaging, optical sensing, clinical diagnosis, and light-emitting displays. However, the proposed method also has some limitations: (1) the ligands require $-NH_2$ groups to form imine bonds; (2) the luminescence enhancement is constrained by the oxidation state of the gold atoms in the core/kernel and the number of surface ligands; and (3) the imine bond is sensitive to pH, and the experimental conditions need to be carefully controlled.

## Methods

**Synthesis of $Au_{22}(SG)_{18}$ NCs**. $Au_{22}(SG)_{18}$ NCs were synthesized according to a previous method with a slight modification[23]. In a typical synthesis, 1 mL $HAuCl_4 \cdot 3H_2O$ (50 mM) and 1.5 mL SG (50 mM) were added to 47.5 mL ultrapure water. After 2 min of vigorous stirring, the pH of the reaction mixture was adjusted to 12.0 using 1 M NaOH. Thereafter, fresh prepared $NaBH_4$ aqueous solution (10 mM, 125 μL) was slowly added to the reaction mixture. The resulting solution was gently stirred for 0.5 h and then the pH was brought to 2.5 using 1 M HCl. After aging for 24 h, the raw product was collected and dialyzed using a dialysis tube (molecular weight cutoff of 3 kDa). To separate $Au_{22}(SG)_{18}$ NCs from the raw product, a native PAGE was carried out. Stacking and separating gels were prepared from 4 and 30 wt% acrylamide/bis-acrylamide, respectively. The raw product (10 μL of gold NC solution in 8 vol% glycerol) was loaded into the wells of the stacking gel. The electrophoresis was allowed to run for 2 h at a constant voltage of 130 V at 4 °C until the three bands were distinctly separated. The red-emitting band was cutoff, crushed, and soaked in ultrapure water for 12 h at 4 °C. The gel lumps suspended in the solution were removed by using membrane filters with 0.22 μm pore size. Finally, the isolated $Au_{22}(SG)_{18}$ NCs was collected and lyophilized into powder.

**Synthesis of PDA-$Au_{22}(SG)_{18}$ NCs**. Typically, 0.2 mL aqueous solution of $Au_{22}(SG)_{18}$ NCs (300 μM) and 0.32 mL aqueous solution of PDA (5 mM) were added to 1.48 mL phosphate buffer (20 mM, pH = 11.0). The mixture was allowed to react for 0.5 h at room temperature and then stored in a 4 °C refrigerator for further use.

**Computational details**. For the density functional theory calculations, all the geometries of reactants, complexes, transition states, intermediates, and products were obtained at the B3LYP/6-311+G(d, p) level by using the Gaussian 16 software. The analytical frequency calculations were computed to identify the nature of stationary points (minima and transition states possess zero and one imaginary frequency, respectively) at the optimized level. Intrinsic reaction coordinate analysis was undertaken to verify the exact connection of the transition states to the desired minima. The used Gibbs free energies are the sum of the single-point energies and the thermal corrections to Gibbs free energy at the temperature of 298 K and pressure of 1 atm. Dimensional plots of molecular configurations and orbitals were generated with the GaussView program.

## Data availability
The authors declare that the data supporting the findings of this study are available within the article and its Supplementary Information, as well as from the corresponding author on request. The Cartesian coordinates for the calculated structures are available within the Supplementary Data 1.

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

## Acknowledgements

We gratefully acknowledge financial support of the National Natural Science Foundation of China (21675024 to W.C. and 21804021 to H.D.), the Program for Innovative Leading Talents in Fujian Province (No. 2016B016 to W.C.), and the Program for Innovative Research Team in Science and Technology in Fujian Province University (No. 2018B033 to W.C.).

## Author contributions

J.X. and W.C. conceived the idea, designed the experiments, and co-supervised this work. H.D., K.H., and L.X. carried out the experiments and characterizations. W.S. performed the DFT calculations. H.D. and K.H. wrote the manuscript. Q.Y., X.F., X.H., H.N., and H.P. discussed the results and commented on the manuscript.

## Competing interests

The authors declare no competing interests.
