## [Peer Review File · Nature Communications]

REVIEWER COMMENTS

Reviewer #1 (Remarks to the Author):

In this manuscript, Deng et al. reported the synthesis, characterization, and intramolecular bi-Schiff base formation in glutathione capped gold nanoclusters. The resulting clusters show enhanced luminescence in the solution state at the single-cluster level.

Materials and methods: The authors prepared three gold nanoclusters (NCs) with glutathione surface ligands viz., Au₂₂(SG)₁₈, Au₁₈(SG)₁₄ and Au₁₅(SG)₁₃, using previously reported procedure. The NCs are well characterized using ESI-mass spectrometry, UV-Vis, fluorescence, and femtosecond transient absorption spectroscopy.

Intramolecular imine bond formation: The authors used dialdehydes to induce imine bond formation between the amine groups of glutathione ligands and the added dialdehydes. Using multiple analytical tools, the authors have shown that the bis-imine bonds are formed between the ligands of the same cluster, i.e., intramolecular, and not intermolecular. Furthermore, pyridine dialdehyde has resulted in a several-fold increase in luminescence among the dialdehydes. The observed increases in luminescence were pronounced for Au₂₂(SG)₁₈ NC. Using time-resolved fluorescence and DFT calculations, the authors have shown that the enhancement in luminescence and quantum yield is correlated with the number of gold atoms in oxidation state, Au(I) and the number of Au atoms to ligand ratio.

The manuscript is logically structured, clearly presented and the hypotheses are supported using multiple experimental and theoretical results. The method provides a new approach to increase the luminescence and quantum yield of molecular clusters without significantly affecting their size and stability.

Relevance and Novelty: Gold nanoclusters are continuously gaining attention due to their unique optoelectronic properties. Due to their photothermal stability, there is a massive interest in using their luminescence properties for various applications including imaging. However, a vast majority of the luminescent NCs display low quantum yield. There have been several approaches to improve the luminescence and quantum yield, either using bulky ligands, aggregation-induced emission, self-assembly enhanced luminescence, and host-guest interactions. Unlike the previously reported reports, the current approach is elegant and straightforward. It does not alter the nanocluster size but increases the luminescence and quantum yield up to ten-fold.

Limitations: The presented method has several limitations. The approach is now shown only for one type of ligand, i.e., glutathione. Furthermore, the ligands require amine groups to achieve imine bond formation. The method has been presented for only gold nanoclusters and enhancement in luminescence is limited by the oxidation state of gold atoms in the core/kernel and the number of surface ligands. Moreover, the observed enhancement is observed only for pyridine dialdehyde. Imine bonds are susceptible to pH and their application, therefore, limited within a narrow range of experimental conditions. Moreover, the observed increase in QY (48%) is still below those observed in Au₈ individual NCs(58.7%) by Jia et al. ACS Nano 2019, 13, 7, 8320.

The manuscript is acceptable to be published after appropriate revision.

Comments and corrections:

1. Line 66, "They can occur Schiff base..." may be a rephrased sentence will be helpful.
2. Line 112 and other parts in the caption, " The yellow balls..." can be replaced with " The yellow spheres.."
3. Line 340 "...a dictahedral Au₉ core" should be "...a dioctahedral core.."
4. Line 303, "...electron spin resonance". Should be "...electron spin resonance spectroscopy"
5. Line 352, in Fig 5, the authors have used Au₂₅(SG)₁₈. However, other characterization data for Au₂₅ NC are missing. Some clarification in this context will be useful.
6. Line 469 and line 496, References 17 and 27 are identical.
7. In Supporting information, on Page S6, Fig. 8, the authors have shown TEM images of Au₂₂(SG)₁₈ and PDA- Au₂₂(SG)₁₈. I wonder how the other aldehydes affected the TEM images of the NCs. Furthermore, how PDA-Au₁₈(SG)₁₄ and PDA-Au₁₅(SG)₁₃
8. Supplementary Figure 9, the authors have shown DLS measurements from 20 nm onwards, which of course, suggest that there is no aggregation. However, the data also doesn't rule out the presence of aggregates below 20 nm. It is helpful to have the DLS with an entire size range. In principle, one could measure particle size of 1 nm. Therefore, please provide a complete spectral region for the benefit of readers.

Reviewer #2 (Remarks to the Author):

The authors describe a strategy to enhance luminescence quantum yield (QY) of water-soluble gold nanoclusters. The paper is well written and most of the conclusions accurately reconcile the data. However, I don't feel this paper is at the level needed for Nature Communications. Various strategies have already been described to enhance the luminescence QY of gold nanoclusters by multiple groups (Refs 14-27). Although the dialdehyde cross-linking strategy has not been used for luminescent gold nanoclusters, it is straightforward to see that the cross-linking of the surface motifs should suppress the nonradiative energy loss and increase the radiative recombination rate. This strategy is also conceptually similar to the author's previous report (Ref 24). The electron dynamics study was well conducted, but the results were mostly the same as those reported previously (Ref 52). My recommendation is to redirect this manuscript to a more specialized journal.

Reviewer #3 (Remarks to the Author):

Atomically precise metal clusters have promising applications in several fields, including biological imaging. For this application the boosting of their quantum yield is of high importance. The authors demonstrate the increase in luminescence quantum yield by linking the molecules in the ligand shell together via a chemical reaction (Schiff base formation). The finding was observed for several clusters.

In general, the discovery is important and I support publication. However, the authors need to clarify some points:

-The pH dependence of the fluorescence would be interesting to know.

-The authors write: ". It can be seen from Fig.3b that the characteristic peak corresponding to the-CHO group at 10.0 ppm almost disappears and a new peak at 8.6 ppm that belongs to the proton of the -CH=N-group is detected upon..." I would say that the first peak is just broad, in contrast to the latter peak, which is very sharp. Probably the integrated intensity of the first peak is much higher compared to the latter.

-The authors write: "Since SG contains only one amino group and each PDA can react with two amino groups to produce bis-Schiff base compound, the binding number of PDA on each Au₂₂(SG)₁₈ cluster is supposed to be 9. This assumption can be confirmed by the PL..." This seems quite speculative to me. Is it not possible to use mass spectrometry to deduce this number?

If indeed all SG ligands have reacted it is highly probable that some PDA has an unreacted -CHO group. As the coverage increases the probability of having two adjacent SG groups (necessary to react with both ends of the PDA molecule) on the cluster surface decreases. This would mean that there are some "dangling" -CHO groups (which could explain the NMR signal). In this case there is the risk of forming aggregates (which could also explain the increased luminescence). The authors use DLS and TEM to exclude this but both measurements do not exclude the formation of small

aggregates. In TEM the particles after treatment seem to be larger (at least some are too large) and the DLS does not consider particle below 20 nm.

Comments by Reviewer #1:

In this manuscript, Deng et al. reported the synthesis, characterization, and intramolecular bi-Schiff base formation in glutathione capped gold nanoclusters. The resulting clusters show enhanced luminescence in the solution state at the single-cluster level.

Materials and methods: The authors prepared three gold nanoclusters (NCs) with glutathione surface ligands viz., $Au_{22}(SG)_{18}$, $Au_{18}(SG)_{14}$ and $Au_{15}(SG)_{13}$, using previously reported procedure. The NCs are well characterized using ESI-mass spectrometry, UV-Vis, fluorescence, and femtosecond transient absorption spectroscopy.

Intramolecular imine bond formation: The authors used dialdehydes to induce imine bond formation between the amine groups of glutathione ligands and the added dialdehydes. Using multiple analytical tools, the authors have shown that the bis-imine bonds are formed between the ligands of the same cluster, i.e., intramolecular, and not intermolecular. Furthermore, pyridine dialdehyde has resulted in a several-fold increase in luminescence among the dialdehydes. The observed increases in luminescence were pronounced for $Au_{22}(SG)_{18}$ NC. Using time-resolved fluorescence and DFT calculations, the authors have shown that the enhancement in luminescence and quantum yield is correlated with the number of gold atoms in oxidation state, Au(I) and the number of Au atoms to ligand ratio.

The manuscript is logically structured, clearly presented and the hypotheses are supported using multiple experimental and theoretical results. The method provides a new approach to increase the luminescence and quantum yield of molecular clusters without significantly affecting their size and stability.

Relevance and Novelty: Gold nanoclusters are continuously gaining attention due to their unique optoelectronic properties. Due to their photothermal stability, there is a massive interest in using their luminescence properties for various applications including imaging. However, a vast majority of the luminescent NCs display low quantum yield. There have been several approaches to improve the luminescence and quantum yield, either using bulky ligands, aggregation-induced emission, self-assembly enhanced luminescence, and host-guest interactions. Unlike the previously reported reports, the current approach is elegant and straightforward. It does not alter the nanocluster size but increases the luminescence and quantum yield up to ten-fold.

Limitations: The presented method has several limitations. The approach is now shown only for one type of ligand, i.e., glutathione. Furthermore, the ligands require amine groups to achieve imine bond formation. The method has been presented for only gold nanoclusters and enhancement in luminescence is limited by the oxidation state of gold atoms in the core/kernel and the number of surface ligands. Moreover, the observed enhancement is observed only for pyridine dialdehyde. Imine bonds are

susceptible to pH and their application, therefore, limited within a narrow range of experimental conditions. Moreover, the observed increase in QY (48%) is still below those observed in Au₈ individual NCs (58.7%) by Jia et al. ACS Nano 2019, 13, 7, 8320.

The manuscript is acceptable to be published after appropriate revision.

Reply: We are greatly encouraged by the reviewer's positive comments on the significance and technical innovation of our work. Indeed, the dialdehyde-mediated cross-linking is a unique and efficient method to enhance the luminescence quantum yield (QY) of gold nanoclusters in aqueous solution, which will open the door for precision surface-engineering on atomically precise nanoclusters and the applications of dynamic bond chemistry. Compared to other approaches, this strategy has several advantages. First, it does not change the size of the nanoclusters, but increases the luminescence intensity (and QY) up to 11-fold. Second, the Schiff base reaction not only enables the obtained products with high stability and good water solubility, but also provides new functions for the obtained materials, which is beneficial to their further applications. Third, the method developed in this study is more general and convenient. For example, a significant luminescence enhancement can be achieved for different types of dialdehydes (m-phthalaldehyde, furan-2,5-dicarbaldehyde, and 2,6-pyridinedicarboxaldehyde) (Fig. 2c) and ligand systems (glutathione, γ -Glu-Cys, Gly-Cys-Gly, and L-penicillamine) (Fig. 2c and Fig. R1). However, as pointed out by the reviewer, the proposed method also has some limitations: 1) the ligands require amine groups to achieve imine bond formation (Fig. R1); 2) the luminescence enhancement is limited by the oxidation state of the gold atoms in the core/kernel and the number of surface ligands; and 3) the imine bond is sensitive to pH and the experimental conditions (Fig. R2) need to be carefully controlled. These limitations have been fully discussed in our revised manuscript.

We really appreciate the helpful and comprehensive comments/suggestions by the reviewer, which have largely improved the readability and scientific contents of our manuscript. These comments/suggestions have been taken into careful consideration in the revised manuscript and a point-to-point response can be found in the following paragraphs.

Fig. R1 (Supplementary Fig. 20 in the revised SI). The effect of 2,6-pyridinedicarboxaldehyde (PDA) on the luminescence of gold NCs protected by (a) γ -Glu-Cys, (b) Gly-Cys-Gly, (c) L-penicillamine, and (d) N-acetyl-L-cysteine (blue line: gold NCs; red line: gold NCs + PDA). Inset shows the chemical structure of the corresponding ligand. Because there is no $-\text{NH}_2$ group to achieve imine bond formation, N-acetyl-L-cysteine-protected gold NCs did not show any significant luminescence improvement with the addition of PDA.

Fig. R2 (Supplementary Fig. 13 in the revised SI). The pH-dependent photoluminescence intensity of PDA-Au₂₂(SG)₁₈ NCs.

Revision:

Page 13, Lines 226-229:

“By contrary, the luminescent intensity was strengthened considerably after being reacted with PDA (Fig. 3d), and this intensity was found to be stable in the pH range of 9 to 12 (Supplementary Fig. 13).”

SI, Page S8, Supplementary Fig. 13:

Fig. R2 is included in SI as Supplementary Fig. 13.

Page 19, Lines 356-359:

“In addition, significant luminescence improvements were also observed for other ligand systems containing $-NH_2$ groups (γ -Glu-Cys, Gly-Cys-Gly, and L-penicillamine) (Supplementary Fig. 20). These results demonstrate the good generality of the as-developed luminescence enhancement strategy.”

SI, Page S11, Supplementary Fig. 20:

Fig. R1 is included in SI as Supplementary Fig. 20.

Page 22, Lines 401-405:

“However, the proposed method also has some limitations: 1) the ligands require $-NH_2$ groups to form imine bonds; 2) the luminescence enhancement is constrained by the oxidation state of the gold atoms in the core/kernel and the number of surface ligands; and 3) the imine bond is sensitive to pH, and the experimental conditions need to be carefully controlled.”

1) Line 66, “*They can occur Schiff base...*” may be a rephrased sentence will be helpful.

Reply: Thanks for your good suggestion. We have rephrased this sentence.

Revision:

Page 4, Lines 69-70:

“Dialdehydes can react with amino groups to form imine ($-CH=N-$) bonds to induce cross-linking of the components, ...”

2) Line 112 and other parts in the caption, “*The yellow balls...*” can be replaced with “*The yellow spheres..*”

Reply: We have changed it accordingly. Thank you.

Revision:

Pages 6 and 20, captions of Figs. 1 and 5:

All “balls” in the text have been replaced with “spheres”.

3) Line 340 “..a dictahedral Au₉ core” should be “..a dioctahedral core..”

Reply: We have changed it accordingly. Thank you.

Revision:

Page 19, Line 346:

“..a dictahedral Au₉ core” has been replaced with “..a dioctahedral Au₉ core..”.

4) Line 303, “..electron spin resonance”. Should be “..electron spin resonance spectroscopy”

Reply: We have changed it accordingly. Thank you.

Revision:

Page 17, Lines 309-310:

“..electron spin resonance” has been replaced with “..electron spin resonance spectroscopy”.

5) Line 352, in Fig 5, the authors have used Au₂₅(SG)₁₈. However, other characterization data for Au₂₅ NC are missing. Some clarification in this context will be useful.

Reply: Thank you for this good suggestion. Au₂₅(SG)₁₈ has been characterized by ESI-MS. This data can be found in Supplementary Fig. 19.

6) Line 469 and line 496, References 17 and 27 are identical.

Reply: We apologize for this mistake and have corrected it accordingly in this revision. Thank you.

7) In Supporting information, on Page S6, Fig. 8, the authors have shown TEM images of Au₂₂(SG)₁₈ and PDA-Au₂₂(SG)₁₈. I wonder how the other aldehydes affected the TEM images of the NCs. Furthermore, how PDA-Au₁₈(SG)₁₄ and PDA-Au₁₅(SG)₁₃.

Reply: Thank you for this insightful comment. As suggested by the reviewer, we have performed TEM measurements on other aldehyde-gold NC systems (Fig. R3). The results showed that no large aggregates or assemblies of gold NCs were generated after interaction with aldehydes, suggesting that the observed photoluminescence enhancement was caused by intracluster cross-linking. These results have been included in our revised manuscript.

Fig. R3 (Supplementary Fig. 9 in the revised SI). TEM images of (a) mPA-Au₂₂(SG)₁₈ and (b) DFF-Au₂₂(SG)₁₈ NCs.

Revision:

Page 12, Lines 210-213:

“The transmission electron microscope (TEM) images demonstrated that the Au₂₂(SG)₁₈ NCs remain individual in the presence of PDA (Supplementary Fig. 8) and other dialdehydes (Supplementary Fig. 9).”

SI, Page S7, Supplementary Fig. 9:

Fig. R3 is included in SI as Supplementary Fig. 9.

8) Supplementary Figure 9, the authors have shown DLS measurements from 20 nm onwards, which of course, suggest that there is no aggregation. However, the data also doesn't rule out the presence of aggregates below 20 nm. It is helpful to have the DLS with an entire size range. In principle, one could measure particle size of 1 nm. Therefore, please provide a complete spectral region for the benefit of readers.

Reply: We appreciate the good suggestion of the reviewer to improve the readability of this figure. As suggested, we have provided DLS spectra with the entire size range (Fig. R4). As can be seen, no aggregates or assemblies of Au₂₂(SG)₁₈ clusters were generated upon interaction with PDA, confirming the intracuster crosslink-enhanced emission by activating RIM at the single-cluster level.

Fig. R4 (Supplementary Fig. 10 in the revised SI). DLS measurements of $\text{Au}_{22}(\text{SG})_{18}$ and PDA- $\text{Au}_{22}(\text{SG})_{18}$ NCs.

Revision:

SI, Page S7, Supplementary Fig. 10:

DLS spectra with an entire size range have been provided.

Comments by Reviewer #2:

The authors describe a strategy to enhance luminescence quantum yield (QY) of water-soluble gold nanoclusters. The paper is well written and most of the conclusions accurately reconcile the data. However, I don't feel this paper is at the level needed for Nature Communications. Various strategies have already been described to enhance the luminescence QY of gold nanoclusters by multiple groups (Refs 14-27). Although the dialdehyde cross-linking strategy has not been used for luminescent gold nanoclusters, it is straightforward to see that the cross-linking of the surface motifs should suppress the nonradiative energy loss and increase the radiative recombination rate. This strategy is also conceptually similar to the author's previous report (Ref 24). The electron dynamics study was well conducted, but the results were mostly the same as those reported previously (Ref 52). My recommendation is to redirect this manuscript to a more specialized journal.

Reply: We are very grateful to the reviewer for his/her efforts in reviewing our manuscript and for providing us with constructive comments and suggestions to further improve the quality of our paper. We are sorry that we might not articulate well the novelty and significance of our work in the previous submission, and we would like to further justify here.

1. Improving the luminescence QY of gold nanoclusters is an interesting topic in the cluster community. It has been reported that the restriction of intramolecular motion (RIM) of surface ligands by solvent- or cation-induced aggregation and self-assembly, enhanced rigidity by binding to bulky groups, host-guest interactions, and spatial confinement, is an efficient approach to enhance the emission efficiency of metal nanoclusters. **However, the above-mentioned systems may have the following drawbacks: (i) they usually generate large-scale architectures with relatively poor stability and controllability; (ii) the resulting products cannot be well dispersed in water; and (iii) surface engineering is often required, resulting in complicated design procedures and poor versatility.** Therefore, there is a pressing need to develop efficient and versatile strategies to enhance the luminescence QY of metal nanoclusters. In our work, a new method was designed to effectively improve the luminescence efficiency of gold nanoclusters by using bis-Schiff base linkages formed between the $-CHO$ groups of dialdehydes and the $-NH_2$ groups of the ligands. **Compared to the reported methods, our strategy has several advantages: (i) it does not change the size of the nanoclusters, but increases the luminescence (and QY) up to 11-fold; (ii) the Schiff base reaction enables the resulting products to possess high stability and good water solubility; (iii) it is more versatile and convenient since amino groups are common functional groups on NC surfaces; and (iv) it will provide the fabricated materials with new functions of Schiff bases (e.g., catalytic activity, antibacterial activity, and reversible luminescence), which is beneficial to their further applications.**

- 2. The dialdehyde cross-linking induced luminescence enhancement of Au₂₂(SG)₁₈ nanoclusters is related to the unique properties of imine bonds.** It was found that the addition of 2,6-pyridinediformyl chloride, which can link the SG ligands with the formation of amide bonds (–CO–NH–), did not increase the luminescence of Au₂₂(SG)₁₈ clusters (Fig. R5). **This data suggests that the observed photoluminescence improvement of the dialdehyde-Au₂₂(SG)₁₈ nanoclusters cannot be simply attributed to the cross-linking of the surface motifs.** Unlike amide bonds, imine bonds can give the fabricated materials a rigid structure, which is beneficial for luminescence enhancement. In addition, the formed bis-Schiff base linkages (imine bonds) reduce the flexibility of the surface-bound Au(I)-SG complexes, thereby shortening the interaction distance between the core and surface motifs of Au₂₂(SG)₁₈ nanoclusters, which can yield a higher T₁ state with a smaller ΔE_{S1-T1}. As a result, the intersystem crossing process in the dialdehyde-Au₂₂(SG)₁₈ NCs becomes much faster than that in Au₂₂(SG)₁₈ NCs, which significantly suppresses the non-radiative energy loss and increases the radiative recombination rate. **Our results demonstrate that the dialdehyde-mediated cross-linking is an efficient way to enhance the luminescence QY of gold nanoclusters, which deepens the understanding of the RIM mechanism, and opens the door for precision surface-engineering on atomically precise nanoclusters and the applications of dynamic bond chemistry.**
- 3.** In our previous work (*Chem. Mater.* 2017, 29, 1362-1369), we reported that the introduction of L-arginine (Arg) into the capping layer of 6-aza-2-thiothymine-protected AuNCs (ATT-AuNCs) can enhance their photoluminescence. The strong host-guest interaction between the guanidine group of Arg and the surface-capped ATT can rigidify the ligand shell and suppress the energy loss processes on the surface of nanoclusters. Although this host-guest interaction-based strategy has been proved as an effective method to enhance the luminescence QY of gold nanoclusters, **the poor versatility and complex design procedures greatly limit its applicability because surface engineering is required to generate a specific host-guest recognition. In addition, the hydrogen bond between the guanidine group of Arg and ATT on the surface of nanoclusters is unstable and can be easily destroyed, which severely hinders its application.** In this study, a new method was designed to effectively improve the luminescence efficiency of gold nanoclusters by using bis-Schiff base linkages formed between the –CHO group of the dialdehydes and the –NH₂ group of the ligands. **Compared with ionic bonds and hydrogen bonds, imine bonds not only possess better stability, but also provide the fabricated materials with new functions of Schiff bases (e.g., catalytic activity, antibacterial activity, and reversible luminescence), which are beneficial to their further applications. Furthermore, this strategy can be readily extended to other ligand systems as –NH₂ groups are common functional groups on nanocluster surfaces (Fig. R6).**

4. $\text{Au}_{25}(\text{SR})_{18}$ and $\text{Au}_{22}(\text{SR})_{18}$ nanoclusters exhibit strong core-shell coupling during their excited-state deactivation (*J. Phys. Chem. C* 2013, 117, 23155-23161; *J. Phys. Chem. Lett.* 2015, 6, 1390-1395). **It is widely believed that these core-shell gold nanoclusters can share a similar electron dynamics model** (*J. Am. Chem. Soc.* 2015, 137, 8244-8250; *J. Phys. Chem. C* 2009, 113, 9440-9444; *J. Phys. Chem. Lett.* 2018, 9, 5303-5310; *Ann. Rev. Phys. Chem.* 2021, 72, 121-142). **The work reported by Moran et al.** (*J. Phys. Chem. C* 2009, 113, 9440-9444) **mainly focused on the construction of the electron dynamics model for $\text{Au}_{25}(\text{SR})_{18}$ nanoclusters, and our study aimed to gain insight into the mechanism of PDA-induced luminescence enhancement of $\text{Au}_{22}(\text{SR})_{18}$ nanoclusters.** By comparing the electron dynamics of $\text{Au}_{22}(\text{SR})_{18}$ and PDA- $\text{Au}_{22}(\text{SR})_{18}$ nanoclusters, some new information that have not been drawn in $\text{Au}_{25}(\text{SR})_{18}$ nanoclusters were obtained: (a) a faster intersystem crossing rate was observed in PDA- $\text{Au}_{22}(\text{SR})_{18}$ nanoclusters, which originates from the unique core-shell structure of $\text{Au}_{22}(\text{SR})_{18}$ nanoclusters (Figs. 4C and 4E); and (b) a slight red shift of the excited-state absorption was found in PDA- $\text{Au}_{22}(\text{SR})_{18}$ nanoclusters, which originates from a higher T_1 state (Figs. 4A and 4D). Furthermore, it was found that $\text{Au}_{22}(\text{SR})_{18}$ nanoclusters display a faster core-shell transition rate than that of $\text{Au}_{25}(\text{SR})_{18}$ nanoclusters (0.3 ps vs 1.4 ps), which leads to a higher luminescence QY (4.6% vs 0.1-0.2%).

From the above justifications, we firmly believe that the novelty and significance of our study meet the requirements of *Nature Communications*. We really appreciate the reviewer's constructive comments/suggestions to further deepen our thinking on this interesting research topic.

Thank you very much.

Fig. R5 The photoluminescence spectra of $\text{Au}_{22}(\text{SG})_{18}$ NCs in the absence (blue line)

and presence (red line) of 2,6-pyridinediformyl chloride.

Fig. R6 The effect of 2,6-pyridinedicarboxaldehyde (PDA) on the luminescence of gold NCs protected by (a) γ -Glu-Cys, (b) Gly-Cys-Gly, and (c) L-penicillamine (blue line: gold NCs; red line: gold NCs + PDA). Inset shows the chemical structure of the corresponding ligand.

Revision:

Page 19, Lines 356-359:

“In addition, significant luminescence improvements were also observed for other ligand systems containing $-\text{NH}_2$ groups (γ -Glu-Cys, Gly-Cys-Gly, and L-penicillamine) (Supplementary Fig. 20). These results demonstrate the good generality of the as-developed luminescence enhancement strategy.”

SI, Page S11, Supplementary Fig. 20:

A supplementary figure has been added to show the generality of the proposed strategy for other ligand systems.

Comments by Reviewer #3:

Atomically precise metal clusters have promising applications in several fields, biological imaging. For this application the boosting of their quantum yield is of high importance. The authors demonstrate the increase in luminescence quantum yield by linking the molecules in the ligand shell together via a chemical reaction (Schiff base formation). The finding was observed for several clusters.

In general, the discovery is important and I support publication. However, the authors need to clarify some points:

Reply: We are glad that the reviewer found our strategy interesting and scientifically significant. Indeed, the dialdehyde-mediated cross-linking is a unique and efficient method to enhance the luminescence quantum yield of gold nanoclusters in aqueous solution, which will open a new door for precision surface-engineering on atomically precise nanoclusters and the applications of dynamic bond chemistry. The detailed comments/suggestions of the reviewer have been fully considered in this revision, and a point-to-point response can be found in the following section. Thank you.

1) *The pH dependence of the fluorescence would be interesting to know.*

Reply: Thank you for this good suggestion. The fluorescence of PDA-Au₂₂(SG)₁₈ was found to be stable in the pH range of 9 to 12 (Fig. R7). At lower pH, the fluorescence becomes weaker due to the hydrolysis of –CH=N– linkages.

Fig. R7 (Supplementary Fig. 13 in the revised SI). The pH-dependent photoluminescence intensity of PDA-Au₂₂(SG)₁₈ NCs.

Revision:

Page 13, Lines 226-229:

“By contrary, the luminescent intensity was strengthened considerably after being reacted with PDA (Fig. 3d), and this intensity was found to be stable in the pH

range of 9 to 12 (Supplementary Fig. 13).”

SI, Page S8, Supplementary Fig. 13:

Figs. R7 is included in SI as Supplementary Fig. 13.

2) *The authors write: “It can be seen from Fig. 3b that the characteristic peak corresponding to the –CHO group at 10.0 ppm almost disappears and a new peak at 8.6 ppm that belongs to the proton of the –CH=N– group is detected upon...” I would say that the first peak is just broad, in contrast to the latter peak, which is very sharp. Probably the integrated intensity of the first peak is much higher compared to the latter.*

Reply: Thank you for this insightful comment and we are sorry for our inappropriate statements. We have rephrased this sentence in our revised manuscript.

Revision:

Page 11, Lines 191-193:

“It can be seen from Fig. 3b that a broad peak at ~10.0 ppm corresponding to the –CHO group and a sharp peak at 8.6 ppm that belongs to the proton of the –CH=N– group were detected in the PDA-Au₂₂(SG)₁₈ clusters⁴⁶.”

3) *The authors write: “Since SG contains only one amino group and each PDA can react with two amino groups to produce bis-Schiff base compound, the binding number of PDA on each Au₂₂(SG)₁₈ cluster is supposed to be 9. This assumption can be confirmed by the PL...” This seems quite speculative to me. Is it not possible to use mass spectrometry to deduce this number?*

Reply: Thank you for this good suggestion. As suggested, we have tried to use HR-ESI to characterize gold nanoclusters with the bis-Schiff base linkage, but we couldn't obtain good signals. The bis-Schiff base reaction occurred under alkaline conditions (pH = 11.0), where NaOH and NH₃•H₂O were used to adjust the pH of the samples. However, the presence of a large amount of Na⁺ and NH₄⁺ would mask the signals of gold nanoclusters and thus we couldn't obtain good ESI-MS of PDA-Au₂₂(SG)₁₈ NCs. In addition, the Schiff base linkages are capable of uncoupling and recoupling dynamically (*Angew. Chem. Int. Ed.* 2019, 58, 9682; *Macromol. Mater. Eng.* 2018, 303, 1800200), which can be easily damaged during ESI-MS measurement. Therefore, it is difficult to accurately measure the PDA binding number. However, post reactions with an amino-reactive reagent suggest that almost all amino groups on the surface of Au₂₂(SG)₁₈ NCs were interacted with PDA, and this observation is consistent with the fluorescence titration experiments. We humbly request the reviewer to consider the difficulty of obtaining the binding number of PDA on each Au₂₂(SG)₁₈ cluster using mass spectrometry, and we believe that the results from UV-vis absorption, photoluminescence, FT-IR, and ¹H NMR

measurements have provided solid evidence for the formation of the bis-Schiff base linkages in gold nanoclusters. Thank you.

4) *If indeed all SG ligands have reacted it is highly probable that some PDA has an unreacted –CHO group. As the coverage increases the probability of having two adjacent SG groups (necessary to react with both ends of the PDA molecule) on the cluster surface decreases. This would mean that there are some “dangling” -CHO groups (which could explain the NMR signal). In this case there is the risk of forming aggregates (which could also explain the increased luminescence). The authors use DLS and TEM to exclude this but both measurements do not exclude the formation of small aggregates. In TEM the particles after treatment seem to be larger (at least some are too large) and the DLS does not consider particle below 20 nm.*

Reply: Thank you for these insightful comments and constructive suggestions. Although the PDA-Au₂₂(SG)₁₈ clusters were purified before ¹H NMR measurement, some unreacted PDA may still be present in the sample. In addition, the Schiff base is a unique type of dynamic covalent bond, i.e., the –CH=N– bond is capable of uncoupling and recoupling dynamically (*Angew. Chem. Int. Ed.* 2019, 58, 9682; *Macromol. Mater. Eng.* 2018, 303, 1800200). Therefore, the weak –CHO signal in the PDA-Au₂₂(SG)₁₈ clusters may originate from the residual PDA or the dissociation of –CH=N– linkages.

At the beginning of the experiments, we have considered the risk of aggregate formation. To avoid intercluster cross-linking, our study used a low concentration of Au₂₂(SG)₁₈ clusters (30 μM). Under this experimental condition, the high local concentration of –NH₂ groups on the surface of Au₂₂(SG)₁₈ clusters would induce rapid intracuster cross-linking by PDA rather than intercluster cross-linking (*ACS Sustain. Chem. Eng.* 2019, 7, 16595-16603). In our revised manuscript, we have provided DLS spectra for the entire size range (Fig. R8). As can be seen, no aggregates or assemblies of Au₂₂(SG)₁₈ clusters were generated upon interaction with PDA, confirming that the intracuster crosslink-enhanced emission is achieved by activating RIM at the single-cluster level. As the ultras-small clusters are sensitive to the high dose electron beam, some of the large particles observed in TEM images could be attributed to the aggregated clusters induced by the electron beam radiation.

Fig. R8 (Supplementary Fig. 10 in the revised SI). DLS measurements of $\text{Au}_{22}(\text{SG})_{18}$ and PDA- $\text{Au}_{22}(\text{SG})_{18}$ NCs.

Revision:

SI, Page S7, Supplementary Fig. 10:

DLS spectra with an entire size range have been provided.

REVIEWERS' COMMENTS

Reviewer #1 (Remarks to the Author):

I read the revised version of the manuscript, reviewer reports, and authors' responses to reviewers' questions.

In the revised version of the manuscript, the authors have addressed the questions raised by the reviewers with appropriate explanations. Further, additional experimental data and appropriate literature references support the authors' comments.

Therefore, the revised version of the manuscript is improved significantly and is acceptable for publication.

Comments/corrections,

In the main text, line 357, the authors state, "In addition, significant luminescence improvements were also observed for

other ligand systems containing –NH₂ groups (γ -Glu-Cys, Gly-Cys-Gly, and L-penicillamine) (Supplementary Fig. 20)." However, the supplementary Fig. 20 shows four ligands, and the captions state "The effect of 2,6-pyridinedicarboxaldehyde (PDA) on the luminescence of gold NCs protected by (a) γ -Glu-Cys, (b) Gly-Cys-Gly, (c) L-penicillamine, and (d) N-acetyl L-cysteine"

Reviewer #3 (Remarks to the Author):

The authors responded to my questions and concerns and changed the manuscript accordingly. I am satisfied although the authors could not completely clarify all points. Also, the fact that at neutral pH the surface modification seems not stable reduces the importance of the work (for biological applications). I am still leaning towards supporting publication of the manuscript.

Comments by Reviewer #1:

I read the revised version of the manuscript, reviewer reports, and authors' responses to reviewers' questions.

In the revised version of the manuscript, the authors have addressed the questions raised by the reviewers with appropriate explanations. Further, additional experimental data and appropriate literature references support the authors' comments. Therefore, the revised version of the manuscript is improved significantly and is acceptable for publication.

Reply: We are glad that the reviewer finds improvements in the revised manuscript, many of which are spurred from the useful comments and suggestions from the reviewer. We would also like to thank the reviewer for review and comments on our manuscript again. All remaining concerns have been taken into careful consideration in this revision, and a point-to-point response to the specific comments could be found in the coming paragraphs below.

Comments/corrections,

In the main text, line 357, the authors state, "In addition, significant luminescence improvements were also observed for other ligand systems containing –NH₂ groups (γ -Glu-Cys, Gly-Cys-Gly, and L-penicillamine) (Supplementary Fig. 20)." However, the supplementary Fig. 20 shows four ligands, and the captions state "The effect of 2,6-pyridinedicarboxaldehyde (PDA) on the luminescence of gold NCs protected by (a) γ -Glu-Cys, (b) Gly-Cys-Gly, (c) L-penicillamine, and (d) N-acetyl L-cysteine"

Reply: We appreciate the reviewer's careful and rigorous attitude towards scientific presentation of data. These results have been stated clearly in our revised manuscript.

Revision:**Page 22, Lines 370-375:**

"In addition, significant luminescence improvements were also observed for other ligand systems containing –NH₂ groups (γ -Glu-Cys, Gly-Cys-Gly, and L-penicillamine) (Supplementary Figs. 20a-20c). Notably, because there is no –NH₂ group to achieve imine bond formation, N-acetyl-L-cysteine-protected gold NCs did not show any obvious luminescence improvement with the addition of PDA (Supplementary Fig. 20d)."

Comments by Reviewer #3:

The authors responded to my questions and concerns and changed the manuscript accordingly. I am satisfied although the authors could not completely clarify all points. Also, the fact that at neutral pH the surface modification seems not stable reduces the importance of the work (for biological applications). I am still leaning towards supporting publication of the manuscript.

Reply: We are very glad to learn that our revisions are satisfactory to the reviewer. We would like to thank the reviewer again for his/her constructive comments and suggestions, which have spurred significant improvements on both the scientific content and readability of this manuscript